biomaterials/biochemistry

Raman spectroscopy, Knoop microhardness, dentine caries, amide I, phosphate

**Author for correspondence:**
A. Banerjee
e-mail: avijit.banerjee@kcl.ac.uk

# Chemo-mechanical characterization of carious dentine using Raman microscopy and Knoop microhardness

M. Alturki[1,3], G. Koller[2,4], U. Almhöjd[1,5] and A. Banerjee[1]

[1]Centre of Oral Clinical Translational Sciences, Faculty of Dentistry, Oral & Craniofacial Sciences, and [2]Centre for Host Microbiome Interactions, Faculty of Dentistry, Oral & Craniofacial Sciences, King's College London, London, UK
[3]Department of Restorative Dental Sciences, College of Dentistry, King Saud University, Riyadh, Saudi Arabia
[4]LCN – London Centre for Nanotechnology, 19 Gordon St, Bloomsbury, London WC1H 0AH, UK
[5]Department of Cariology, Institute of Odontology, The Sahlgrenska Academy, University of Gothenburg, Box 450, SE-405 30 Gothenburg, Sweden

MA, 0000-0003-0973-7212; AB, 0000-0003-0091-7348

One of the aims in the clinical operative management of dental carious lesions is to remove selectively the highly infected and structurally denatured dentine tissue, while retaining the deeper, repairable affected and intact, healthy tissues for long-term mechanical strength. The present study examined the correlation of chemical functional groups and the microhardness through the different depths of a carious lesion using Raman spectroscopy and Knoop microhardness testing. The null hypothesis investigated was that there was no correlation between Raman peak ratios (amide I : phosphate $_{v1}$) and equivalent Knoop microhardness measurements. Ten freshly extracted human permanent teeth with carious dentine lesions were sectioned and examined using high-resolution Raman microscopy. The ratio of absorbency at the amide I and phosphate bands were calculated from 139 scan points through the depth of the lesions and correlated with 139 juxtaposed Knoop microhardness indentations. The results indicated a high correlation ($p < 0.01$) between the peak ratio and the equivalent Knoop hardness within carious dentine lesions. This study concluded that Raman spectroscopy can be used as a non-invasive analytical technology for *in vitro* studies to discriminate the hardness of carious dentine layers using the peak ratio as an alternative to the invasive, mechanical Knoop hardness test.

# 1. Introduction

In the contemporary minimally invasive (MI) approach to the operative caries management of active cavitated carious lesions, it is now recommended to remove selectively only the superficial, highly bacterially contaminated and denatured zone of tissue that is clinically wet, soft and sticky—the caries-infected dentine. The remaining deeper caries-affected tissues can be healed and repaired by the dentine-pulp complex and therefore can be retained and sealed off using bio-interactive restorative materials. This surgical procedure preserves more dental tissue and ultimately improves the long-term survival of the dentine-pulp complex. The transition between these two histologically different zones of tissue, however, is rather diffuse and difficult to identify clinically, which leads to subjectivity between operators and therefore, often unnecessary excessive removal of tooth tissue. Thus, the delineation of the endpoint of carious dentine excavation of the infected or contaminated zone is essential, both clinically and in laboratory investigations [1–7].

The most common carious dentine management techniques are mechanical in nature, ranging from diagnostics (dental probing of carious dentine) through to surgical intervention (using hand or rotary instruments combined with tactile clinical hardness feedback as the discriminator between the zones). Several laboratory studies have therefore focused on the mechanical properties of carious dentine using relative changes in tissue hardness, measured using Knoop hardness, as a gold standard to determine the characteristics and borders of these different carious dentine histological zones [8–10]. However, the hardness test has a major drawback in that it is an invasive, low-resolution test that damages samples, often precluding any further analysis of the tissues [11].

Recently, the biochemical tissue changes occurring during the caries process have gained significant research interest, as these insights aid understanding of the pathophysiology of lesion development during the caries process, and this knowledge can assist development of approaches to help remineralize/heal such compromised dental tissues as opposed to surgically excising them and replacing them with biologically inadequate artificial biomaterials [12,13]. Raman spectroscopy has emerged in the dental field as a non-invasive technology for *in vitro* studies, after sectioning the teeth, to characterize chemically both healthy and carious dentine, through the characteristic molecular vibrational energy signatures [14]. Several studies have used Raman spectroscopy to describe the mineral and matrix peak distributions across a carious dentine lesion, representing the inorganic and organic components of the carious tissues [15–19]. Appreciating the detailed chemical changes within the carious dentine tissues is important for the comparison and development of new selective minimally invasive operative management of carious dentine removal techniques, to identify potential naturally occurring biomarkers to help delineate the necrotic from repairable tissue and to develop novel research into further chemical adhesion mechanisms for bio-interactive restorative materials.

This study aimed to determine the chemical functional correlation, using non-invasive Raman spectroscopic analysis, of the normalized amide I:phosphate peak intensity ratios in carious dentine with invasive mechanical Knoop microhardness measurements. The null hypothesis investigated was that there was no correlation between Raman peak ratios (amide I:phosphate $_{v1}$) and Knoop microhardness measurements.

# 2. Material and methods

## 2.1. Sample preparation for combined Raman and hardness measurements

Ten teeth with active carious dentine lesions having an international caries detection and assessment system (ICDAS) lesion score less than 4 [20] were collected and placed in distilled water and stored in a cold cabinet (+4°C) prior to sectioning. Samples were hemi-sectioned longitudinally using a slow-speed (200 r.p.m.) water-cooled diamond blade (Diamond wafering blade XL 12205, Benetec Ltd, UK) and images of the cut surfaces captured using a digital camera (lens: Nikkor AF-S Micro 60 mm f/2.8G ED, D800E, Nikon, Japan). In order to guarantee a flat surface for accurate measurements, the cut surfaces were polished using a polishing machine (MetaServ 3000, Buehler, USA) with silicon waterproof abrasive papers sequentially (P1200 for 10 s, P2500 for 10 s and P4000 for 4 min). A small steel round rotary burr was used to create a fiducial dot reference on each sample (figure 1A). The fiducial mark in each sample was visible using Raman spectroscopy and the Knoop microhardness indenter (KHN). The distance between the reference point and the examined points in each sample was measured in micrometres (µm) (figure 1B). Thus, we can identify the exact point in both tests.

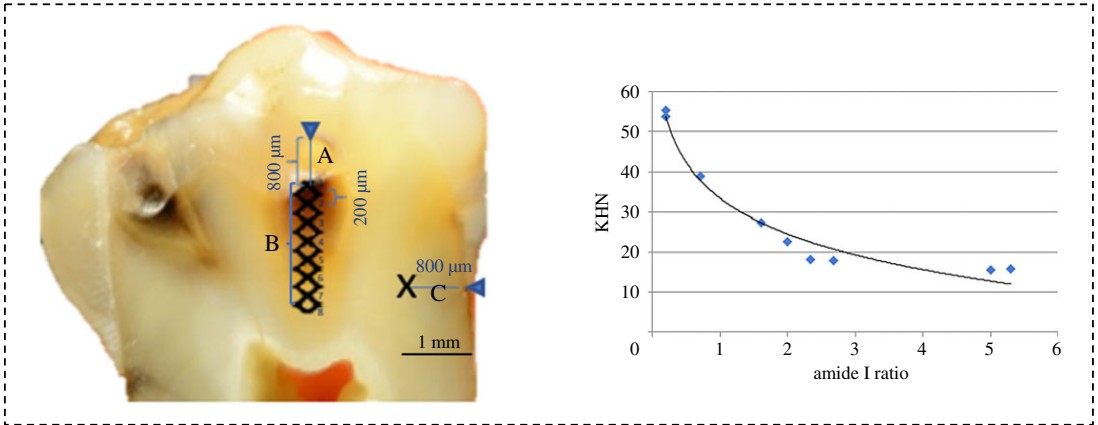

**Figure 1.** A representative diagram for one of the examined samples shows: (A) enlarged cross section of the tooth with carious dentine after creating the fiducial marker 800 µm from the occlusal pit. (B) the eight examined points under Raman microscopy and Knoop hardness tester, (C) The sound dentine examined point under KHN and Raman microscopy, and a scatter plot with a regression line (R) of the hardness on the peak ratio (amide I : phosphate $_{v1}$).

The prepared lesion hemi-sections were firstly analysed using the non-invasive Raman microscope and then using the KHN microhardness indenter, at the same measurement locations that originated from the enamel–dentine junction through the lesion in a straight path with control measurements taken on areas of sound dentine in the same sample, figure 1.

## 2.2. Raman spectroscopy

A high-resolution Raman microscope (inVia, Renishaw Plc, Wotton-under-Edge, UK) running in Streamline scanning mode was used to scan the tooth surfaces. A total of 139 point scans were taken across the 20 carious dentine samples. A 5/0.12 NA air objective was used. In each sample, the distance between the fiducial mark and the points scanned were measured in micrometres via the Raman software, figure 1B. Spectrum acquisition was conducted using a 785 nm laser (0.5 mW laser power) and a 600 line mm$^{-1}$ grate over the spectral band width of 4000–400 cm$^{-1}$ for each sample. The point scanning time was around 2 min per sample. Baseline correction was performed by the Raman processing software (WiRe, Renishaw, UK). The distance between each point scan was 200 µm.

## 2.3. Knoop microhardness

A Struers Duramin KHN microhardness indenter (Struers Ltd, Denmark) was used to measure the microhardness values of the dentine at the same 139 points measured using Raman spectroscopy. For each microhardness measurement, KHN diamond-shaped indenter was placed into the central point of each Raman analysing point, i.e. the single spectrum (figure 1B). A 10 g load was used for 15 s to produce a diamond-shaped indentation that was measured using a 40/0.65 NA objective in triplicate. The manufacturer's software (ecos workflowTM v. 2.16.0, 2007–2016 emco-test prüfmaschinen gmbh, Austria) was used to calculate the KHN for each point.

## 2.4. Statistical analysis and spectral correlation processing with hardness measurements

Raman peaks at 1650 and 960 cm$^{-1}$ were used for estimating the matrix to mineral ratio earlier described [21] as the relationship between demineralized and mineralized tissue. About 1650 cm$^{-1}$ represents the CO-stretch of the amide I band (CONH) and 960 cm$^{-1}$ represents the phosphate vibrations, respectively. After baseline correction in single spectra, the peak ratios were calculated by dividing the intensity of amide I with the intensity of phosphate; ($I_{1650}/I_{960}$ (cm$^{-1}$/cm$^{-1}$)), figure 2. The correlation of matrix-mineral ratio and Knoop microhardness (KHN) at each indentation was calculated using logarithmic regression analysis (SPSS 25 software, IBM, USA).

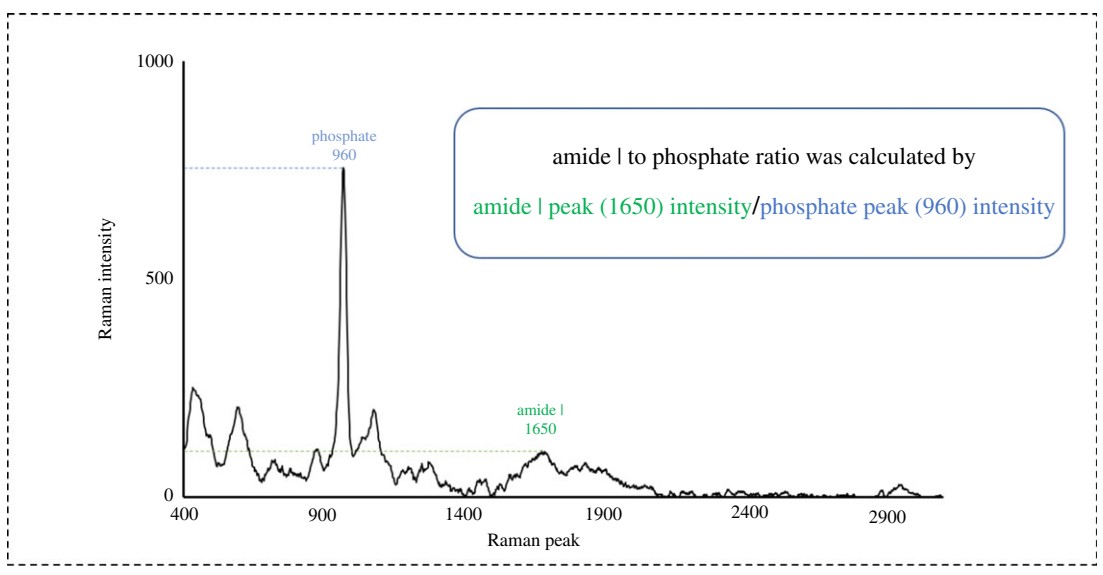

**Figure 2.** Representative Raman peaks chart for one of the examined samples after normalization by processing software (WiRe, Renishaw, UK), explaining the methodology of calculation the peak ratio (amide I : phosphate $_{v1}$) in each scan point.

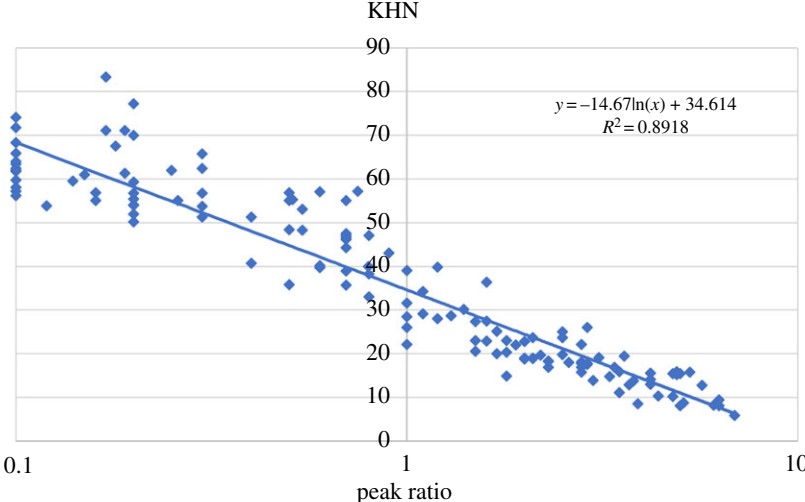

**Figure 3.** A scatter plot and a regression line ($R$) of the microhardness (KHN) vs the peak ratios (amide I : phosphate $_{v1}$) in the 20 carious dentine samples, illustrating the de novo prediction of histological status, including hardness of the tissue across the carious lesion. It shows the difference between reduced mineral (more amide 1), greater than 1 peak ratio, and more mineral (more phosphate $_{v1}$), less than 1 peak ratio.

# 3. Results

## 3.1. Raman Spectral correlation with Knoop microhardness

The 139 examined points from 20 samples of carious dentine lesions were analysed with Raman spectroscopy and Knoop microhardness tester, to correlate the peak ratio (amide I : phosphate $_{v1}$) with the Knoop hardness numbers, figure 1.

The peak ratio values (amide I : phosphate $_{v1}$) and the Knoop hardness numbers (KHN) in 20 samples of carious dentine lesions were plotted as a scatter diagram, and the coefficient of determination was calculated across the samples as illustrated in figure 3, $R^2 = 0.89$ (multiple $R = 0.89$, adjusted $R^2 = 0.89$ and s.e. = 6.67) with $p$-value less than 0.001, which indicating a significantly high correlation between peak ratio and KHN.

KHN and peak ratio shows a logarithmic regression relationship. The KHN ($y$-axis) corresponding to the peak ratio ($x$-axis) could be formularized by a method of $R$ square as $y = -14.67\ln(x) + 34.61$ with a significant $p$-value less than 0.001 and the 95% CI (33.48–35.74). Using this formula (regression line), the hardness of carious dentine would be calculated when the peak ratio was measured.

# 4. Discussion

In conventional operative dentistry, the process of carious tissue removal is based on complete removal of tooth tissue affected by the caries process, as it was originally thought that the lesion represented the disease process and therefore to prevent spread of the disease, the lesion had to be fully excised to sound tissue. This was thought also to provide a mechanically strong substrate prior to restoration but led to more tooth loss and ultimately catastrophic failure of the tooth-restoration complex. In recent years, with improved understanding of the caries process and biomaterials research, it is now appreciated that the repairable caries-affected, or even some infected dentine may be retained during operative intervention [4–7,22,23]. However, the excavation procedure in the clinic remains an essentially mechanical process, be it with rotary instruments or by tactile discrimination of excavation corresponding to the respective microhardness values. Thus, the *in vitro* correlate to these removal procedures is the relative microhardness of caries-infected, affected and healthy tooth structure, and this measure remains the gold standard of *in vitro* characterization of carious lesions. This represents a direct surrogate measure of the hydroxyapatite (HA) crystals present in the structure, thereby making this a valid comparator and correlate to mechanical excavation.

Carious dentine lesions have been characterized mechanically in many studies. Ogawa *et al.* [24] described using microhardness to characterize carious dentine. Hardness increased gradually from the outer to the inner surface of the lesion. Other studies classified the carious dentine into two zones according to their hardness values, caries-infected dentine (CID) less than 25 and caries-affected dentine (CAD) between 25 and 40 [11,25]. The CID layer has lower KHN value because of greater dissolution of HA compared with the deeper CAD layers [26,27]. Thereby, hardness values have become a global validation to evaluate the effectiveness of mechanical caries removal techniques [5,10,12,23,28]. The data from this study regarding the levels of microhardness agrees with previous studies in that the hardness increases from superficial heart of the lesion towards the inner (deeper) lesion surface of carious dentine.

The relation between KHN and other diagnostic technologies such autofluorescence (AF) was studied [11,29]. They found a statistically significant negative correlation between the AF signal intensity of carious dentine and its relative microhardness. This meant that as the tissue hardened, i.e. became healthier, the AF signal decreased significantly [11,29].

From a biochemical perspective, the peaks of phosphate and collagen amide I have been used in spectroscopic studies of calcified tissues to analyse a chemical distribution in carious dentine lesions. The phosphate peak is the most intense Raman peak in dental hard tissues and is found most intensively in sound enamel and dentine, where minerals are the main content [30]; amide I peak is the most prominent component of the organic content in dentine [18]. Almahdy *et al.* [29] used Raman microscopy to describe the mineral proportion represented by the phosphate (P–O) peak at 960 cm$^{-1}$. They showed that this analytical technology could highlight the significant mineral reduction in infected dentine compared with sound dentine [29]. Moreover, Maske et al. [31] used FTIR to analyse a developed biofilm cariogenic challenge model and showed that caries-affected dentine had a reduced amide I content compared with sound dentine, since lower amide I peak intensities were observed in the CAD [31]. The band ratio between phosphate and amide I was discussed in different studies [15,16]. Liu *et al.* [16] used the peak ratio to describe the mineral transitions in affected carious dentine from the transparent zone (TZ) into the normal zone (NZ), as they compared the phosphate to amide I peaks, using FTIR. They found that the mineral is high within the sub-transparent zone compared with normal and transparent zones [16]. However, Almhöjd *et al.* [15] used the peak ratio (amide I : phosphate) to analyse the amide I distributions in carious dentine and showed that when comparing the amide I band with the phosphate band in both sound dentine and carious dentine, an interesting difference in the ratio was observed, with lower values for the carious dentine compared with sound dentine. This means that the intensity of amide I of dentine proteins is higher in carious dentine than sound dentine [15], which corresponds to the present Raman peak ratio result, as the peak ratio decreased from infected carious dentine towards sound tissue.

The present study compared Raman peak ratios and their equivalent KHN measurements in carious dentine. Previous studies have associated the KHN with the relative mineral content of dental hard tissues, either to identify the histological changes of the different carious dentine layers [16] or to test new applied restorative materials [28], because mineral content is used routinely as a reference for the inherent physical properties of hard tissues [32,33]. However, the mathematical correlation between a carious dentine peak ratio (amide I : phosphate $_{v1}$) and their KHN measurements has not been identified before. According to a statistical analysis of the presented results, a high correlation percentage has

founded between KHN and Raman peak ratio. Therefore, the null hypothesis investigated in this study was rejected as the correlation coefficient $R^2 = 89\%$ across all tested samples. Thus, Raman peak ratio (amide I : phosphate $_{v1}$) can be used to discriminate the physical hardness of the carious dentine zones.

# 5. Conclusion

Raman spectroscopy proved to be a non-invasive *in vitro* diagnostic technology, using the peak ratio (amide I : phosphate $_{v1}$) as an alternative to the invasive, low-resolution Knoop hardness test. This diagnostic technique has the potential to improve the *in vitro* evaluation of carious dentine non-destructively. Moreover, it could be used to help find the optimum carious dentine removal technique that reduces the risk of removing unnecessary tissue during cavity preparation. Further work should explore the protein structural data obtained by Raman to indicate if the denaturation and arising clinical utility can be modelled/characterized by Raman microscopy to develop and improve MI approaches to caries.

Ethics. Ten extracted human teeth with cavitated carious dentine lesions were collected using an ethics protocol reviewed and approved by NHS Health Research Authority (16/SW/0220).

Data accessibility. Data are available from the Dryad Digital Repository: https://dx.doi.org/10.5061/dryad.qjq2bvqcf [34].

Authors' Contributions. M.A., K.G., U.A. and A.B. contributed to conception, design, data acquisition, analysis and interpretation, drafted and critically revised the manuscript. All authors gave final approval and agree to be accountable for all aspects of the work.

Competing interests. The authors declare no conflict of interests in relation to this study.

Funding. The authors acknowledge the College of Dentistry, King Saud University, Riyadh, Saudi Arabia for grant sponsorship of the first author (grant no. 4/52/141590 dated 13/12/2018). The funder had no role in study design, data collection and analysis, decision to publish or preparation of the manuscript.

Acknowledgements. The authors gratefully acknowledge the assistance rendered by Mr Peter Pilecki in the microscopical analysis. In addition, the authors would like to gratefully acknowledge Mrs Fiona Warburton for her assistance in the statistical analysis.

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
