## [Reviewer comments · Royal Society Open Science]

Review History

RSOS-200404.R0 (Original submission)

Review form: Reviewer 1

Is the manuscript scientifically sound in its present form?

Yes

Are the interpretations and conclusions justified by the results?

Yes

Is the language acceptable?

Yes

Do you have any ethical concerns with this paper?

Yes

Have you any concerns about statistical analyses in this paper?

No

Recommendation?

Accept as is

Comments to the Author(s)

No comments

Review form: Reviewer 2

Is the manuscript scientifically sound in its present form?

Yes

Are the interpretations and conclusions justified by the results?

Yes

Is the language acceptable?

Yes

Do you have any ethical concerns with this paper?

No

Have you any concerns about statistical analyses in this paper?

No

Recommendation?

Accept as is

Comments to the Author(s)

Thank you for inviting me to comment on Chemo-Mechanical Characterisation of Carious Dentine Using Raman Microscopy and Knoop Microhardness. I see that the paper has already been thoroughly reviewed and all queries and suggestions fully dealt with prior to it being transferred to this journal. I reviewed the paper, the reviewers' comments and authors' changes and believe they will have satisfied the reviewers. don't have anything further to add and think the paper should be published as it is (with the change of burr to bur - which is more usual in British English).

Decision letter (RSOS-200404.R0)

Dear Dr Alturki:

It is a pleasure to accept your manuscript entitled "Chemo-Mechanical Characterisation of Carious Dentine Using Raman Microscopy and Knoop Microhardness." in its current form for publication in Royal Society Open Science. The comments of the reviewer(s) who reviewed your manuscript are included at the foot of this letter.

on behalf of Professor Luning Liu (Associate Editor) and Dr Pietro Cicuta (Subject Editor).

Associate Editor Professor Luning Liu Comments to Author:

Thanks for submitting the manuscript and responses to previous reviewers' comments. The manuscript has been reviewed by other two reviewers and both are happy with the scientific content and changes. Therefore, I am pleased to accept the manuscript. Please pay attention to the minor suggestion of reviewer 2 during the proof correction.

Reviewer(s)' Comments to Author:

Reviewer: 1

Comments to the Author(s)

No comments

Reviewer: 2

Comments to the Author(s)

Thank you for inviting me to comment on Chemo-Mechanical Characterisation of Cariou Dentine Using Raman Microscopy and Knoop Microhardness. I see that the paper has already been thoroughly reviewed and all queries and suggestions fully dealt with prior to it being transferred to this journal. I reviewed the paper, the reviewers' comments and authors' changes and believe they will have satisfied the reviewers. don't have anything further to add and think the paper should be published as it is (with the change of burr to bur - which is more usual in British English).
